# Design, Synthesis and Biological Evaluation of Novel PEG-Rakicidin B1 Hybrid as *Clostridium difficile* (CD) Targeted Anti-Bacterial Agent

**DOI:** 10.3390/molecules28166152

**Published:** 2023-08-21

**Authors:** Lijun Xie, Li Chen, Yongbo Wei, Nannan Chen, Tong Wu, Jingming Zhou, Hong Jiang, Feng Lin

**Affiliations:** Fujian Provincial Key Laboratory of Screening for Novel Microbial Proucts, Fujian Institute of Microbiology, Fuzhou 350007, China; chenli821228@163.com (L.C.); 13110774889@163.com (N.C.); qitianfeng@gmail.com (T.W.); jemin_1011@126.com (J.Z.); jianghong709@163.com (H.J.)

**Keywords:** Rakicidin B1, PEGylation strategy, *anti-Clostridium difficile*, low cytotoxicity

## Abstract

Rakicidin B1 was isolated and purified from the culture broth of a marine *Streptomyces* sp. as a potent anti-cancer agent, and lately the compound and its derivatives have firstly been found to possess anti-*Clostridium difficile* (CD) activity but with high cytotoxicity. Herein, following our previous discovery on anti-CD activity of Rakicidin B1, structure modification was performed at the OH position of Rakicidin B1 and a new Rakicidin B1-PEG hybrids FIMP2 was facilely designed and synthesized by conjugating the PEG2000 with the scaffolds of Rakicidin B1 via the linkage of carbamate. The cytotoxicity of the FIMP2 was first evaluated against three different cancer cell lines, including HCT-8 cells, PANC-1, and Caco-2, with IC_50_ values at 0.519 μM, 0.815 μM, and 0.586 μM, respectively. Obviously, as compared with a positive control group treated with Rakicidin B1, the IC_50_ value of FIMP2 increased by nearly 91-fold, 50-fold, and 67-fold, suggesting that the PEGylation strategy significantly reduced the cytotoxicity of FIMP2. Thus, this preliminary result may be beneficial to increase its safety index (SI) value due to the decreased cytotoxicity of FIMP2. In addition, this decreased cytotoxicity of FIMP2 was further confirmed based on a zebrafish screening model in vivo. Thereafter, the anti-CD activity of FIMP2 was evaluated in vivo, and its efficacy to treat CDI was found to be better than that of vancomycin. The mortality and recurrence rate of FIMP2 is not as low compared with that of vancomycin; these results demonstrated that compound FIMP2 is a new, promising anti-CD agent with significant efficacy against CD recurrence with low cytotoxicity towards bodies.

## 1. Introduction

As a Gram-positive spore-forming anaerobic bacterium, *Clostridium difficile* (CD) infection can cause illness ranging from diarrhea, colitis, and serious toxic intestinal conditions to death [1,2]. Normally, *C. difficile* infection (CDI) is mainly triggered by the use of antibiotics, which are harmful to the reproduction of normal microflora, allowing CD to proliferate in the colon and induce CDI [3]. Recently, the number of CDIs has been growing, causing an enormous medical problem. In 2017 alone, CDI resulted in 223,900 hospitalizations and 12,800 deaths in the United States, with nearly $6.3 billion in estimated healthcare costs [4]. Therefore, the development of new, efficient, therapeutic strategies to fight against CD is meaningful to address CDI.

Currently, CDI is treated predominately with metronidazole (MTZ), vancomycin (VAN), and fidaxomicin (FDX) (Figure 1) [5]. Metronidazole and vancomycin are recommended for the treatment of mild to moderate episodes with a high response in approximately 90% of patients. However, recurrence of CDI has high rates of approximately 25–30% [6,7] due to the loss of beneficial gut microbiota in patients. In view of the transient efficacy of metronidazole and vancomycin, fidaxomicin has been shown to lower recurrence compared with vancomycin in clinical trials, but it still faces the challenge of recurrence and potent cytotoxic effects [8,9,10]. More seriously, since the outbreak of the hypervirulent strain, such as the BI/NAP1/PCR ribotype 027 strain, the incidence of CDIs continues to trend upward, and the efficacy of these antibiotics has been challenged [11,12,13]. Therefore, new antibacterial agents that selectively target CD are still highly desirable to tackle persistent and recurrent CDI.

Fortunately, some progress has been made in recent years. For example, a new anti-CDI agent called ridinilazole is currently undergoing clinical trials for the treatment of CDI; it remains to be seen whether it would provide any benefit over current treatments. In our previous work, the Rakicidin analogues have been characterized and identified as promising anti-CDI agents [14]. Next, a series of Rakicidin B1 derivatives were synthesized and obtained as new potent candidates and have exhibited encouraging results [15,16]. Despite their obvious value as a new type of anti-CDI agent, their therapeutic value is limited, to a great extent, due to their cytotoxicity and poor solubility. Thus, to overcome these shortcomings, it is still necessary to search for and discover novel, soluble Rakicidin analogues with anti-CDI activity and lower cytotoxicity, which will increase the space and possibility for us to develop Rakicidin analogues as new anti-CDI agents.

As an extension of our work on the discovery of novel potent CD inhibitors based on the Rakicidin B1 scaffold, herein, the PEGylation strategy has been employed successfully to construct target FIMP2 by attaching the strands of the polymer PEG2000 to the scaffold of Rakicidin B1. Next, the resulting compound FIMP2 was evaluated against three human cancer cell lines and a zebrafish screening model. More significantly, in vivo investigation demonstrated that FIMP2 has more potent efficacy to treat CDI in mice with a decreased recurrence rate and mortality rate as compared with that of vancomycin. Taken together, FIMP2 may have the potential to be developed as a new anti-CDI agent with low systematic cytotoxicity.

## 2. Results

### 2.1. Chemistry

The synthetic route of the target compound FIMP2 was illustrated in Figure 1; the key intermediate (Z-01) was synthesized from Rakicidin B1 by treatment with 4-(chloromethyl) benzoyl chloride in a pyridine solution at −5 °C for 0.5 h, and then subsequently reacted with sodium azide under KI as a catalyst to afford the key intermediate compound Z-02 as white powder. Next, the key intermediate was reacted with alkyne-modified polyethylene glycol (MPEG2000-ALK) to afford the crude target compound FIMP2. The products obtained were purified by column chromatography on silica gel. The chemical structures of all the synthesized novel compounds were confirmed by ^1^HNMR and HRMS spectra (Appendix A). Furthermore, the chemical structure of the target product FIMP2 was confirmed by ^13^CNMR (Appendix A).

### 2.2. In Vitro Cytotoxicity of FIMP2

In order to study the toxic effect of FIMP2 compounds on intestinal cells in the treatment of CD, three gastrointestinal cancer cell lines were selected for in vitro toxicity tests. As shown in Figure 2, the in vitro cytotoxicity of FIMP2 and Rakicidin B1 was evaluated by the (Cell Counting Kit-8) CCK-8 assay. All the results expressed as IC_50_ (concentration of the compound producing 50% cell growth inhibition after 48 h of drug exposure) are summarized in Appendix A. Under normoxia conditions, Rakicidin B1 exhibits a potent inhibitory impact on HCT-8 cells, PANC-1 and Caco-2, with IC_50_ values at 0.519 μM, 0.815 μM, and 0.586 μM, respectively. However, the IC_50_ value of FIMP2 increased by nearly 91-fold, 50-fold, and 67-fold in each of the three cancer cells under the same conditions, suggesting that the PEGylation strategy significantly reduced the cytotoxicity of FIMP2. Furthermore, cytotoxicity tests were carried out in hypoxia environments to simulate the hypoxia environment in the colon. The cytotoxicity of FIMP2 was reduced by 66~79-fold compared to that of the parent compound Rakicidin B1. In a word, compound FIMP2 modified by PEG2000 showed higher IC_50_ values, presenting low cytotoxic potential in all three cell lines, both under normoxia and hypoxia conditions.

### 2.3. In Vivo Cytotoxicity of FIMP2

To obtain more robust evidence of reduced cytotoxicity of compounds, a zebrafish pancreatic cancer transplant model was selected to evaluate the in vivo cytotoxicity of FIMP2. In this experiment, PANC-1 cells were labeled with red fluorescent dye, and then a PANC-1 transplantation model was obtained by microinjection. Because zebrafish have optically transparent tissues [17,18], the cytotoxicity could be evaluated by analyzing the fluorescence intensity of tumor cells. In the pancreatic cancer model, gemcitabine hydrochloride was selected as the positive control. As shown in Table 1 and Figure 3, the positive control group showed a significant inhibition rate (42%) at a drug concentration of 66.7 µM, indicating that the screening is workable. Under the same conditions, when the dosage of compound Rakicidin B1 gradually increased from 22.4 nM to 201.5 nM, the fluorescence intensity of tumor cells showed a consequent concentration-dependent decrease, suggesting that the growth of tumor cells was inhibited, and the tumor inhibition rate (43%) at this time was comparable to that of the positive control group, indicating that compound Rakicidin B1 exhibited significant cytotoxicity. However, when the PEG-modified target FIMP2 was administered at doses of 10.6 µM and 31.9 µM, the fluorescence intensity of tumor cells was not significantly reduced, and there was no significant difference in the analysis of statistical results. When the dose reached the minimum toxic concentration (MTC, 95.8 µM), it showed a weak tumor cells inhibition rate (30%), suggesting that the introduction of PEG2000 significantly reduced the cytotoxicity of FIMP2 as compared with parent compound Rakicidin B1.

### 2.4. In Vivo Efficacy of FIM-P2 in a CDI Mouse Model

Based on its low cytotoxicity, we further investigated whether FIMP2 retained the potential anti-CD activity of Rakicidin B1. We adopted the established in vivo CD mouse model of Reeves et al. [19] in which mice were initially treated with cefoperazone sodium and clindamycin, and then infected with CD to induce CDI. As shown in Figure 4, mice were infected and treated for seven days, and then they were monitored for survival and possible CD recurrence until the 21st day. In this model, half of the Rakicidin B1-treated and vehicle-treated mice died after three days of infection. Although none of the vancomycin-treated mice died, the mortality rate reached 40% within two days of recurrence of infection, which showed a similar mortality rate (50%) compared to Rakicidin B1-treated and vehicle-treated mice. However, only 20% of FIMP2-treated mice died during treatment, indicating that FIMP2 was effective in reducing mortality from CDI. Vancomycin-treated mice survived the first seven days, as previously reported [20,21], but 40% of vancomycin-treated mice died five days after stopping vancomycin treatment. In contrast, FIMP2 significantly protected mice from CDI recurrence with 100% survival after a seven-day treatment period (Figure 4).

These conclusions could be further demonstrated by body weight changes and clinical scores [22] in infected mice (Figure 5). The vancomycin-treated mice did not experience weight loss or an increase in clinical scores while receiving treatment, demonstrating that vancomycin showed significant antibacterial properties. However, the mice showed a significant weight loss and a sharp increase in clinical scores after stopping vancomycin treatment, indicating the occurrence of CDI recurrence [23]. Conversely, weight loss and increased clinical scores occurred in the FIMP2 treatment group from the first day to the fifth day after infection; however, mortality did not reappear, and clinical scores decreased significantly after stopping treatment, indicating that a recurrence of infection did not occur after treatment ended. These results showed that compound FIMP2 is a potent anti-CD agent with significant efficacy against CD recurrence.

## 3. Materials and Methods

### 3.1. Synthesis

#### 3.1.1. General Procedure

Reagents and solvents (all anhydrous HPLC-grade) were obtained from commercial suppliers and used without any further purification unless otherwise stated. All reagents were weighed and handled in air unless otherwise stated. Reactions were monitored by using thin layer chromatography (TLC) by means of Macherey-Nagel silica gel 0.20 mm (60-F_254_) under UV light (l = 254 nm). High resolution mass spectra (HRMS) were recorded on a GCT premier CAB048 mass spectrometer operating in the MALDI-TOF mode. ^1^H NMR spectra were recorded on a Bruker Avance-600/400 MHz spectrometer. Chemical shifts (δ) were reported in ppm and were calibrated to the residual signals of the deuterated solvent (DMSO-*d_6_*).

#### 3.1.2. Synthesis of Intermediate Z-01

Rakicidin B1 (0.5 g, 0.8 mmol) and pyridine (10.0 mL) were added to a 50-mL eggplant flask, and 4-(chloromethyl) benzoyl chloride (0.76 mL, 5 mmol) was slowly added dropwise under −5 °C agitation. The reaction was completed by TLC detection after about 0.5 h after dropping. Then, a large amount of water was added until a light pink insoluble substance precipitated, and after filtration, the filter cake was placed in a vacuum drying oven and baked for 1 h at 50 °C for later use. HRMS (ESI): calcd for C_41_H_61_ClN_4_O_8_ [M + H]^+^ 773.4211, C_41_H_61_ClN_4_O_8_; found, [M + H]^+^ 773.4250, [M + Na]^+^ 795.4072 (Appendix A). ^1^H NMR spectroscopic data as shown in Appendix A. ^1^H NMR (400 MHz, DMSO-*d*_6_): δ 9.01 (s, 1H), 8.65 (d, *J* = 10.3 Hz, 1H), 8.20 (d, *J* = 8.0 Hz, 2H), 7.85 (s, 1H), 7.61 (d, *J* = 8.0 Hz, 2H), 7.30 (s, 1H), 6.92 (d, *J* = 14.9 Hz, 1H), 6.21 (d, *J* = 14.9 Hz, 1H), 5.56–5.44 (m, 3H), 5.34 (m, 1H), 5.13 (d, *J* = 10.4 Hz, 1H), 4.87 (s, 2H), 4.61 (d, *J* = 18.1 Hz, 1H), 3.83 (d, *J* = 18.2 Hz, 1H), 3.00 (s, 3H), 1.25 (m, 18H), 1.12 (q, *J* = 7.0 Hz, 6H), 1.04 (d, *J* = 7.0 Hz, 6H), 0.89 (d, *J* = 6.8 Hz, 3H), 0.86 (m, 2H), 0.83 (d, *J* = 6.3 Hz, 6H).

#### 3.1.3. Synthesis of Intermediate Z-02

In a 100-mL eggplant flask, the intermediate Z-01 (1 g, 1.29 mmoL) was added to a DMF solution containing NaN_3_ (0.487 g, 7.5 mmoL), KI (0.096 g, 0.58 mmoL). During the process, HPLC (Agilent C18 column, 5 m, 250 mm 4.6 mm, methanol-water 850:150, column temperature 40 °C, flow rate 1.0 mL/min, wavelength 262 nm) was used to track and stop the reaction. After 0.5 h at 55 °C, the reaction solution was poured into 200 mL of saturated saline, light yellow-brown insoluble matter was precipitated, filtered, and washed with water and ether respectively, and the filter cake was dried and directly used for the next reaction. HRMS (ESI): calcd for C_41_H_61_N_7_O_8_ [M + H]^+^ 780.4615, C_41_H_61_N_7_O_8_; found, [M + H]^+^ 780.4637, [M + Na]^+^ 802.4460 (Appendix A). ^1^H NMR spectroscopic data as shown in Appendix A. ^1^H NMR (600 MHz, DMSO-*d*_6_) δ 8.97 (s, 1H), 8.62 (d, *J* = 10.1 Hz, 1H), 8.22 (d, *J* = 8.1 Hz, 2H), 7.82 (s, 1H), 7.54 (d, *J* = 8.0 Hz, 2H), 7.27 (s, 1H), 6.92 (d, *J* = 14.8 Hz, 1H), 6.19 (d, *J* = 15.0 Hz, 1H), 5.49 (s, 1H), 5.44 (d, *J* = 8.4 Hz, 2H), 5.32 (s, 1H), 5.12 (m, 1H), 4.60 (m, 3H), 3.81 (d, *J* = 18.2 Hz, 1H), 2.99 (s, 3H), 1.29 (m, 2H), 1.24 (s, 11H), 1.12 (d, *J* = 8.5 Hz, 4H), 1.09 (d, *J* = 9.0 Hz, 2H), 1.04 (d, *J* = 7.0 Hz, 2H), 0.95 (m, 2H), 0.90 (d, *J* = 6.6 Hz, 1H), 0.87 (d, *J* = 6.8 Hz, 3H), 0.85 (d, *J* = 1.4 Hz, 3H), 0.84 (d, *J* = 3.0 Hz, 3H), 0.83 (d, 6H).

#### 3.1.4. Synthesis of Intermediate FIMP2

The intermediate Z-02 (0.56 g, 0.72 mmoL) was added to a DMF solution of alkyne-modified polyethylene glycol MPEG2000-ALK (1.0 g), a small amount of CuSO_4_·5H_2_O (50 mg, 0.2 mmol), and sodium ascorbate (100 mg, 0.5 mmol) in a 100-mL eggplant flask. The reaction mixture was stirred at ambient temperature for 2 h, and the reaction was completely tracked and stopped using HPLC (Agilent C18 column, Agilent Technologies, Santa Clara, CA, USA, 5 μm, 250 mm × 4.6 mm, methanol–water 850:150, column temperature 40 °C, flow rate 1.0 mL/min, wavelength 262 nm) during the reaction. After the end of the reaction, the collected reaction solution was separated by YMC ODS C18 column, 50-μm, 190-mm × 60.8-mm semi-preparative liquid chromatography, with a volume ratio of 830:170 methanol–water as the eluent. The flow rate of the eluate was controlled by 15 mL/min for elution, collected in segments, concentrated, and freeze-dried; that is, the final product polyethylene glycol-Rakicidin B1 prodrug derivative FIMP2 was obtained. The average molecular weight was confirmed by HRMS to be 2610.5395 (Appendix A). ^1^H and ^13^C NMR spectroscopic data are shown in Appendix A.

### 3.2. Method Details of Cytotoxicity Studies

#### 3.2.1. In Vitro Cytotoxicity Analysis

A panel of several types of immortalized human cancer cell lines—including pancreas (PANC-1), colon (HCT-8), and colon glands (Caco-2)—were used to test the cytotoxicity of B1 and FIMP2 and associated control compounds. The cells were grown in normal cell culture conditions at 37 °C under a 5% CO_2_ humidified atmosphere, either in DMEM medium for PANC-1, Caco-2 and RPMI-1640 medium for HCT-8. The 100-μL cell suspension (PANC-1 4.0 × 10^4^/mL, HCT-8, Caco-2 5.0 × 10^4^/mL) was then seeded into a 96-well plate. The test compounds were diluted in corresponding medium and added to the cells in a volume of 100 μL. A negative control group was created concurrently. The 96-well cell culture plates were incubated in a 37 °C, 5% CO_2_ normoxia incubator and a 37 °C, 5% CO_2_, 95% N_2_ hypoxic incubator for 48 h. The cytotoxicity was determined using the CCK-8 (KGM12800, Jiangsu Keygen Biotech Co., Ltd., Nanjing, China) colorimetric assay. Optical density (OD) was quantified by spectrophotometry at λ = 450 nm (EL×800, Bio-Tek Instruments, Winooski, VT, USA). IC_50_ values were calculated by using the GraphPad Prism 9.0.0 software. The IC_50_ values are the average of at least three independent experiments. The apparent permeability inhibition rate of the tested agents was calculated using the equation below.
$$\mathrm{inhibition}\;\mathrm{rate}\;\left(\%\right)=\frac{\mathrm{OD}\left(\mathrm{Negative}\;\mathrm{control}\right)-\;\mathrm{OD}(\mathrm{Experimental})}{\mathrm{OD}(\mathrm{Negative}\;\mathrm{control})}\times 100\%$$

#### 3.2.2. In Vivo Cytotoxicity Analysis

Zebrafish were all reared in fish culture water at 28 °C (water quality: 200 mg of instant sea salt was added per 1 L of reverse osmosis water, and the conductivity was 450~550 μS/cm; pH was 6.5~8.5; hardness was 50~100 mg/L CaCO_3_). The experimental animal use license number was SYXK (Zhejiang) 2012-0171. Feeding management meets the requirements of the international AAALAC certification (certification number: 001458). Wild-type AB strain zebrafish were reproduced in natural pairs with 30 individuals per experimental group and an age of 2 dpf. Red fluorescently labeled pancreatic cancer cells (200 cells/fish) were transplanted into the yolk sac of zebrafish by microinjection (Microinjector: PCO-1500, Zgenebio, Taipei, Taiwan) to establish a human pancreatic cancer transplantation model in zebrafish. B1 and FIMP2 as well as positive control medicines were infused into water and cultured for 48 h in the zebrafish pancreatic cancer model after it had reached 3 dpf at 35 °C. Each experimental group chose 10 fish to photograph under a fluorescence microscope (motorized focused continuous zoom fluorescence microscopy: AZ100, Nikon, Tokyo, Japan) and then used NIS-Elements D 3.20 sophisticated image processing software to gather information to analyze tumor fluorescence intensity (S) and assess cytotoxicity using the results of this index’s statistical analysis.
$$\mathrm{inhibition}\;\mathrm{rate}\;\left(\%\right)=\frac{S\;\left(\mathrm{Model}\;\mathrm{control}\right)-\;S\;(\mathrm{Experimental})}{S\;\left(\mathrm{Model}\;\mathrm{control}\right)}\;\times \;100\%$$

#### 3.2.3. CDI Mouse Model

In this experiment, the highly pathogenic CD strain ATCC 43255 (VPI 104643, Shanghai WuXi AppTec New Drug Development Co., Ltd., Shanghai, China) was used for infection. Vancomycin (Sigma-94747, Sigma Aldrich (Shanghai) Trading Co., Ltd., China), cefoperazone sodium (Aladdin-C136445, Aladdin Reagent (Shanghai) Co., Ltd., China), and clindamycin (Sigma-C5269, Sigma Aldrich (Shanghai) Trading Co., Ltd., China) were purchased commercially. C57BL/6 female mice aged 7 weeks were purchased from Shanghai Slac Laboratory Animal Co., Ltd. (Shanghai, China). The study was reviewed, approved, and performed following the guidelines of the Institutional Animal Care and Use Committee (IACUC). Seven-week-old female pathogen-free C57BL/6 mice were pretreated with cefoperazone sodium in sterile drinking water for ten days to disrupt the mice’s normal intestinal microflora, facilitating infection with the toxigenic strain of CD. The mice were given regular water after two days of feeding and then given a single dose of clindamycin (10 mg/kg) intraperitoneally one day prior to the CD challenge. The following day, the mice were infected intragastrically with CD (0.2 mL, 8.4 × 10^6^ CFU/mL). Four hours postinfection, mice were given a single oral dose of compound FIMP2 at 125 mg/kg twice a day, Rakicidin B1 at 25 mg/kg twice a day, vancomycin at 20 mg/kg once a day, or vehicle (0.05%Tween-80 + 0.5%CMC-Na) twice a day. Treatments were continued for seven days, and mice were closely monitored for disease signs (including weight changes, clinical scores, survival curves) for 21 days.

## 4. Conclusions

In our continued efforts to develop more potent rakicidin analogues as anti-CD agents, an efficient strategy of PEGylation was employed to construct target FIMP2, which was synthesized by going through a 3-step reaction. Next, its activity against cancer cell lines in vitro was investigated, and FIMP2 was found to have reduced cytotoxicity, which may be advantageous for minimizing systemic toxicity in the body. Furthermore, its low cytotoxicity was further confirmed by using zebrafish pancreatic cancer transplant models. Importantly, FIMP2, as a new hybrid of Rakicidin B1 with PEG2000, showed much more potent activity anti-CDI with a decreased recurrence rate (0%) and mortality rate (20%) as compared with the positive control group treated with vancomycin. This result demonstrated that FIMP2 is particularly attractive to be developed as a new class of anti-CDI agent. Moreover, the decreased cytotoxicity of FIMP2 may possibly be caused by the embedding of the PEG2000 group. Conversely, hydrolysis of carbonate, freeing Rakicidin B1 at specific lesion locations, should be responsible for its improved CDI efficacy in vivo.

## Data Availability

Data will be subject to availability on request to the corresponding author.

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
