# Peer review of "Design, Synthesis and Biological Evaluation of Novel PEG-Rakicidin B1 Hybrid as Clostridium difficile (CD) Targeted Anti-Bacterial Agent"

_molecules, 2023, doi:10.3390/molecules28166152_

Round 1

Reviewer 1 Report

The work with title “Design, Synthesis and Biological Evaluation of Novel PEG-Rakicidin B1 Hybrid as Clostridium difficile (CD) Targeted Anti-bacterial Agent ” is a valuable and suitable contribution to be published in the Molecules journal after justifying some points.

The provided work describes the isolation and purification of Rakicidin B1, a compound obtained from a marine Streptomyces sp, which has shown promising potential as an anti-cancer agent. However, the researchers discovered that Rakicidin B1 and its derivatives also possess anti-Clostridium difficile (CD) activity, but they exhibit high cytotoxicity, which may limit their therapeutic use.

The researchers hypothesize that this decrease in cytotoxicity could potentially increase the safety index (SI) of FIMP2, making it a safer option for therapeutic use. To validate the reduced cytotoxicity, the study also used a zebrafish screening model in vivo, which confirmed the decreased toxicity of FIMP2. However, all of these data support this work’s impact. but there are some major and minor issues should be resolved

·       the similarity rate is high especially in the methods section and some parts of the introduction which should be reduced accordingly.

·       Unify the Anti-CD in the whole MS

·       Line 68 in vivo should be in italic style

·       Some aromatic and non-aromatic cycles in figure 1 of vanco. needs clean up after drawing.

·       In the caption of scheme 1 no need to re-write 4-(chloromethyl) benzoyle chloride since you draw it in the scheme

·       Line 93 In vitro in italic

·       Line 95 edit And

·       I would like to ask the authors that why they used a cancer cell lines instead of Normal cell lines ?? to evaluate the toxicity ??

·       Line 107-114 the footnote should be summarized and the written paragraph there should be moved to the method section

·       Some data of figure 2 are mentioned and written in the table 1 so update the figure

·       According the results of the In vivo, as we are talking about a molecule with confirmed molecular weight, I would like to see the concentrations with µM-nM instead of g/ml, the Table 2 with different units of concertation make the readers confusing, so edit and correct these issues accordingly

·        Line 185-191 add the information regarding the used HRMS instrument

·       All chemical formula should not be written like C41H61ClN4O8 the numbers should be subscript as C41H61ClN4O8

·       where is the data of the 13C-NMR ???

·       the Conclusion should be improved

Best wishes

Author Response

Thanks a lot for your work on our manuscript entitled “Design, Synthesis and Biological Evaluation of Novel PEG-Rakicidin B1 Hybrid as Clostridium difficile (CD) Targeted Anti-bacterial Agent” (Manuscript ID: molecules- 2557999). Your and reviewers’ comments are very helpful for improving the quality of this manuscript. We have studied the comments carefully and made corresponding corrections as follows. (In the manuscript called “molecules-2557999-revised+manuscript+marked”, we mark the revised text in a yellow background. In the manuscript called “molecules-2557999-revised+manuscript”, as a final version, there is no any mark)

Reviewer #1

Comments: The work with title “Design, Synthesis and Biological Evaluation of Novel PEG-Rakicidin B1 Hybrid as Clostridium difficile (CD) Targeted Anti-bacterial Agent” is a valuable and suitable contribution to be published in the Molecules journal after justifying some points.

The provided work describes the isolation and purification of Rakicidin B1, a compound obtained from a marine Streptomyces sp, which has shown promising potential as an anti-cancer agent. However, the researchers discovered that Rakicidin B1 and its derivatives also possess anti-Clostridium difficile (CD) activity, but they exhibit high cytotoxicity, which may limit their therapeutic use.

The researchers hypothesize that this decrease in cytotoxicity could potentially increase the safety index (SI) of FIMP2, making it a safer option for therapeutic use. To validate the reduced cytotoxicity, the study also used a zebrafish screening model in vivo, which confirmed the decreased toxicity of FIMP2. However, all of these data support this work’s impact. but there are some major and minor issues should be resolved.

Response: Thanks a lot for your comments. We have revised our article according to your comments and suggestions.

Comment 1: the similarity rate is high especially in the methods section and some parts of the introduction which should be reduced accordingly.

Response: Thanks for your suggestions. We have reduced accordingly the similarity rate in the methods section and introduction.

Comment 2: Unify the Anti-CD in the whole MS

Response: Thanks for your suggestions. We have unified the Anti-CD in the whole MS.

Comment 3: Line 68 in vivo should be in italic style

Response: Thanks for your careful checking. We have corrected it in line 68 and in the whole MS.

Comment 4: Some aromatic and non-aromatic cycles in figure 1 of vanco. needs clean up after drawing.

Response: Thanks for your careful checking. We have redrawn the structure in figure 1.

Comment 5: In the caption of scheme 1 no need to re-write 4-(chloromethyl) benzoyle chloride since you draw it in the scheme

Response: Thanks for your suggestions. We have deleted it in the caption of scheme 1.

Comment 6 and 7: Line 93 In vitro in italic; Line 95 edit And.

Response: Thanks for your careful checking. We have corrected it in line 94, 95 and in the whole MS.

Comment 8: I would like to ask the authors that why they used a cancer cell lines instead of Normal cell lines ?? to evaluate the toxicity ??

Response: Thanks for your question. In our previous experiments, we found that the toxicity of FIMP2 to cancer cell lines was significantly reduced, and Caco-2 was widely used to evaluate the toxicity of compounds in the intestine. Thanks for your constructive advice, we will use normal cells to evaluate their toxicity later.

Comment 9: Line 107-114 the footnote should be summarized and the written paragraph there should be moved to the method section

Response: Thanks for your suggestions. We resummarized the data, and the corresponding paragraph has been removed. The written paragraph “The IC50 values are the average of at least three independent experiments” has been moved to the method section (line 247).

Comment 10: Some data of figure 2 are mentioned and written in the table 1 so update the figure

Response: Thanks for your suggestions. We have moved Table 1 to Support information (Table S1). And we have updated the figure 2.

Comment 11: According the results of the In vivo, as we are talking about a molecule with confirmed molecular weight, I would like to see the concentrations with µM-nM instead of g/ml, the Table 2 with different units of concertation make the readers confusing, so edit and correct these issues accordingly

Response: Thanks for your suggestions. We have corrected these issues in the whole MS.

Comment 12: Line 185-191 add the information regarding the used HRMS instrument

Response: Thanks for your reminder. We have added the information of HRMS instrument in line 190-191.

Comment 13: All chemical formula should not be written like C41H61ClN4O8 the numbers should be subscript as C41H61ClN4O8

Response: We are sorry for bothering you with this kind of mistakes that we should have avoided. We have revised them.

Comment 14: where is the data of the 13C-NMR ???

Response: Thanks for your question. In this project, we obtained FIMP2 through directed synthesis based on the well-known compound Rakicidin B1. Therefore, we did not perform 13C-NMR testing of the intermediates in the experiment, but we confirmed the structure by 1H-NMR and HRMS, and the structure of the target compound FIMP2 was confirmed by 13C-NMR and 1H-NMR. Thanks for your understanding.

Comment 15: the Conclusion should be improved

Response: Thanks for your suggestions. We have revised and updated the conclusion section.

Once again, thank you very much for your constructive comments and suggestions, which would help us both in English and in depth to improve the quality of the manuscript.

Kind regards.

Lijun Xie

Fujian Institute of Microbiology

Fuzhou, 350007, China

Phone: +86-150-05008196

Fax: +86-591-8344-1193

Reviewer 2 Report

The quality of images must need to be improved.

The cell densities used in the cell viability tests are in an unusually high range for 96 well plates. Please, explain.

What about 13C-NMR of Z-01? Upload it in the supplementary materials.

References should be ordered according to the mdpi rules.

Minor English grammatical errors need to be identified and resolved. 

Author Response

Thanks a lot for your work on our manuscript entitled “Design, Synthesis and Biological Evaluation of Novel PEG-Rakicidin B1 Hybrid as Clostridium difficile (CD) Targeted Anti-bacterial Agent” (Manuscript ID: molecules- 2557999). Your and reviewers’ comments are very helpful for improving the quality of this manuscript. We have studied the comments carefully and made corresponding corrections as follows. (In the manuscript called “molecules-2557999-revised+manuscript+marked”, we mark the revised text in a yellow background. In the manuscript called “molecules-2557999-revised+manuscript”, as a final version, there is no any mark)

Reviewer #2

Comment 1: The quality of images must need to be improved.

Response: Thanks for your suggestions. The images in our manuscripts are all 300 dpi, which are clearly visible in the word version, but it is possible that the images are automatically compressed when exporting as the PDF version.

Comment 2: The cell densities used in the cell viability tests are in an unusually high range for 96 well plates. Please, explain.

Response: Thanks for your question. The cell densities have been supplemented in methods section (in line 239). And the cell densities should be within the normal range.

Comment 3: What about 13C-NMR of Z-01? Upload it in the supplementary materials.

Response: Thanks for your question. In this project, we obtained FIMP2 through directed synthesis based on the well-known compound Rakicidin B1. Therefore, we did not perform 13C-NMR testing of the intermediates in the experiment, but we confirmed the structure by 1H-NMR and HRMS, and the structure of the target compound FIMP2 was confirmed by 13C-NMR and 1H-NMR. Thanks for your understanding.

Comment 4: References should be ordered according to the mdpi rules.

Response: Thanks, we have ordered the references according to the rules of the MDPI.

Once again, thank you very much for your constructive comments and suggestions, which would help us both in English and in depth to improve the quality of the manuscript.

Kind regards.

Lijun Xie

Fujian Institute of Microbiology

Fuzhou, 350007, China

Phone: +86-150-05008196

Fax: +86-591-8344-1193

Round 2

Reviewer 1 Report

All requested comments were resolved accordingly the MS was improved Well.

Best wishes